# Optimization of Key Hydraulic Structure Parameters of a New Type of Water–Pesticide Integrated Sprinkler Based on Response Surface Experiment

**Junping Liu** [1,*], **Xinjian Wang** [1], **Qingsong Liu** [1], **Zawar Hussain** [1,2] **and Yuxia Zhao** [1]

[1] Research Centre of Fluid Machinery Engineering and Technology, Jiangsu University, Zhenjiang 212013, China

[2] Department of Agricultural Engineering, Bahauddin Zakariya University, Multan 60800, Pakistan

\* Correspondence: liujunping401@163.com

**Abstract:** To meet the requirements of trellis grape crop root irrigation, spraying pesticides on branches and leaves, an integrated sprinkler was designed, which relies on the flow pressure to change the irrigation water and spray pesticide working modes. The structural parameters that affect the hydraulic performance were selected based on the working principle of the sprinkler. The key parameters for the irrigation mode included diversion hole inclination angle, refractive cone angle, refractive cone length, and cone hole distance. The key parameters for the spray pesticide mode included diversion chute width, the number of diversion chutes, the diversion chute inclination angle, the rotary acceleration chamber height, and the nozzle outlet cylindrical section length. The central composite design response surface tests of the water–pesticide integrated sprinkler were carried out; the analysis of variance and regression analysis were selected; the main influence rules and interactions of key structural parameters on irrigation performance and pesticide spraying performance of sprinkler irrigation system were obtained. The optimal parameters of the water–pesticide integrated sprinkler were: the diversion hole inclination angle is 20.8°, the refractive cone angle is 123.7°, the refractive cone length is 8.8 mm, the cone hole distance is 3.6 mm, the diversion chute width is 2.5 mm, the number of diversion chutes is 2, the diversion chute inclination angle is 10°, the rotary acceleration chamber height is 1.3 mm, and the nozzle outlet cylindrical section length is 0.7 mm. The irrigation hydraulic performance of the wetted radius is 3.4 m, the average irrigation application rate is 0.65 mm/h, and the uniformity coefficient is 88%. The spraying pesticide performance of the droplet volume mid-diameter is 200.2 μm, the droplet spectrum width is 2.2, and the droplet coverage is 9.4%.

**Keywords:** water–pesticide; sprinkler; hydraulic performance; irrigation; pesticide application

## 1. Introduction

In recent years, China has increased its fruit production significantly [1]. In 2019, China's grape planting area was 726.2 hectares, accounting for 5.9% of the total orchard planting area [2]. Most of grapes are planted using hedgerow trellis or pergola trellis. The current irrigation methods for trellised grapes are drip irrigation or micro-sprinkler irrigation. The pesticides spraying methods are mainly manual and sprayer. In the process of spraying pesticides with a spray machine, a large amount of pesticides enter the air, which not only wastes pesticides, but also pollutes the environment [3,4]. Manual spraying is time consuming and labor intensive, and due to prolonged contact with pesticides by sprayers, it is easy to cause poisoning and death [5,6]. These methods of applying pesticides also have some problems, such as low pesticide utilization and pollution of the orchard environment [7,8]. Therefore, new application methods need to be explored.

An effective way to solve the problem of excessive utilization of water and pesticides in the current agricultural production process is to combine the functions of irrigation and

pesticide application [9,10]. Water–pesticide integration technology have been applied to drip irrigation. The specific practice is to deliver the desired chemicals to the root for reducing diseases of the roots and improving the effective utilization of pesticides [11,12]. However, the common diseases of grapes are mainly in the leaves or branches of crops [13]. The water–pesticide integrated sprinkler irrigation technology directly acts on the surface of crop leaves through the scattering and atomization of water flow. This can take into account the irrigation water and the branches spraying function [14]. This requires that the sprinkler can meet both irrigation and spraying pesticide performance. At present, the commonly used sprinklers for fruit trees are centrifugal sprinklers, fan atomizing sprinklers, and pneumatic electrostatic sprinklers, among others [15–17]. They do meet the requirement of spraying pesticide on the branches and leaves of crops, but can not meet the requirement of irrigating the roots of crops. Therefore, it is a great significance to develop a dual function sprinkler that can spray pesticides on the branches and irrigate the roots simultaneously.

At present, there are few pieces of research on the water–pesticide integrated sprinkler which can realize the function of water–pesticide co-operative operation in the same sprinkler irrigation system. In 2021, Zhang Qing et al. [14] designed a water–pesticide integrated sprinkler, in which the influence law of key structural parameters on the performance of irrigation and pesticide spraying was studied. Their results provided a theoretical for the design and development of a water–pesticide integrated sprinkler. The grape irrigation cycle water demand is large and needs a sprinkler with a large flow rate. At the same time, the quantity of grape pesticide spraying liquid is small and needs a sprinkler with a small flow rate. However, this sprinkler has only one nozzle; there is a problem of unreasonable spraying flow. In 2022, Wang et al. [18]. developed a water–pesticide integrated sprinkler; the sprinkler needs to adjust the spraying mode manually. The multifunctional aspect of the sprinkler needs to be improved to the automatic form.

In this paper, a new type of water–pesticide integrated sprinkler was developed. It can meet the two working modes in the same system, with irrigation water at low pressure and spraying pesticide at high pressure. The influence of the main structural parameters on the performance of irrigation and pesticide spraying were explored. The optimal structural parameters were obtained.

## 2. Materials and Methods

### 2.1. Structural Design and Working Principle

The structural design of the water–pesticide integrated sprinkler is shown in Figure 1. The test prototype is shown in Figure 2.

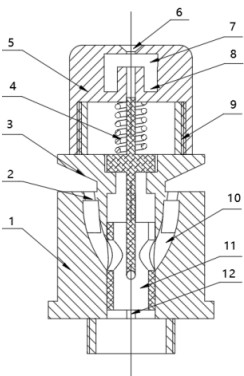

**Figure 1.** Schematic of the new type ofwater–pesticide integrated sprinkler structure. 1. Spinkler body; 2. Deflection hole; 3. Refractive cone; 4. Compression spring; 5. Spinkler cap; 6. Spray nozzle; 7. Rotary acceleration chamber; 8. Diversion chute; 9. Thread; 10. Runner; 11. Valve spool; 12. Square slot.

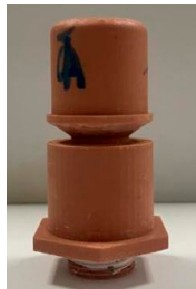

**Figure 2.** Experiment prototype.

　　The working principle of the sprinkler is as follows: In low-pressure irrigation operation (100~200 kPa), the water pressure on the valve spool is less than the elastic force of the spring. Water from the diversion hole encountered the refraction cone, forming a thin fan-shaped layer of water on both sides of the jet. Under air resistance, the water breaks into small droplets and falls on the ground, completing the irrigation function. In medium and high spraying pesticide operation (100~200 kPa), the pressure on the valve spool is greater than the spring elasticity. The valve spool moves up to close the diversion hole so that the liquid can only flow into the diversion chute. The liquid is rotated into the cavity through the diversion chute and is ejected from the nozzle in the shape of conical mist after vortex motion is generated on the cavity wall. Small droplets of atomized liquid are sprayed upward, acting on the back of the crop leaves, to complete the pesticide spraying function.

　　Select the key structure that affects the sprinkler's irrigation and spraying performance, as shown in Figure 3.

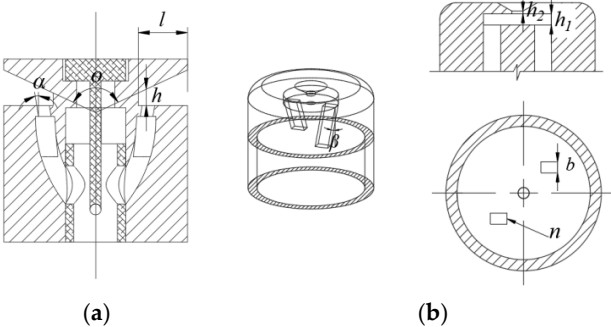

(**a**)　　　　　　　　　　　(**b**)

**Figure 3.** The key hydraulic structure of the sprinkler. (**a**) Structure parameters of irrigation, (**b**) structure parameters of spraying pesticides. $\alpha$: the diversion hole inclination angle; $\theta$: the refractive cone angle; $h$: the cone hole distance; $l$: the refraction cone length; $b$: the diversion chute width; $n$: the number of diversion chutes; $\beta$: the diversion chute inclination angle; $h_1$: the rotary acceleration chamber height; $h_2$: the nozzle outlet cylindrical section length.

　　Figure 3a shows the structure parameters associated with irrigation: the diversion hole inclination angle $\alpha$, the refractive cone angle $\theta$, the refraction cone length $l$, and the cone hole distance $h$ were selected; these structures were impacted on irrigation performance. The upper and lower limits value were: the diversion hole inclination angle is 10–20°, the refraction cone length is 7–11 mm, the refractive cone angle is 120–140°, and the cone hole distance is 2.5–4.5 mm.

　　Figure 3b shows the structure parameters associated with spraying pesticide: the diversion chute width $b$, the number of diversion chute $n$, the diversion chute inclination angle $\beta$, the rotary acceleration chamber height $h_1$, and the nozzle outlet cylindrical section length $h_2$ were selected; these structures were impacted on pesticide spraying performance. The upper and lower limits were: the diversion chute width is 1–2 mm, the number of diversion chute is 2–4, the diversion chute inclination angle is 10–30°, the rotary acceler-

ation chamber height $h_1$ is 2–4 mm, and the nozzle outlet cylindrical section length $h_2$ is 0.1–0.5 mm.

### 2.2. Experimental Setup

The experiments were carried out in the Sprinkler Irrigation Laboratory of Jiangsu University (Zhenjiang, China). The irrigation performance system is shown in Figure 4 and spraying pesticide performance is shown in Figure 5. The equipments for tests included pipeline, centrifugal pump (CM1-7, Grund fu Corp., Shanghai, China), pressure gauge (precision class is 0.4, Hongqi Corp., Xi'an, China), electromagnetic flowmeter (precision class is 0.5, Shunlaida Corp., Xi'an, China), sprinkler, measuring tap, and catch cans. With the sprinkler as the center, four lines with an angle of 30° were laid out for measuring data. The diameter of the catch can is 20 cm, the interval of catch cans is 0.5 m [19–21]. When the system working pressure was 200 kPa, the function for irrigation has been worked and the test time is 30 min. Thirty-one groups of tests were carried out [22]. The wetted radius, average irrigation application rate, and uniformity coefficient were selected to evaluate the sprinkler irrigation performance [14].

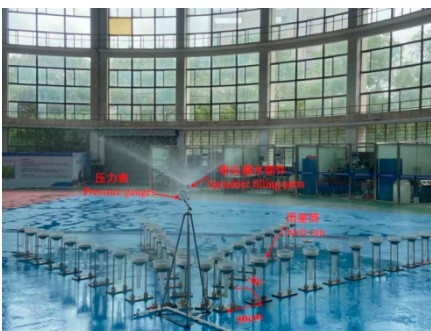

**Figure 4.** Layout of irrigation water performance.

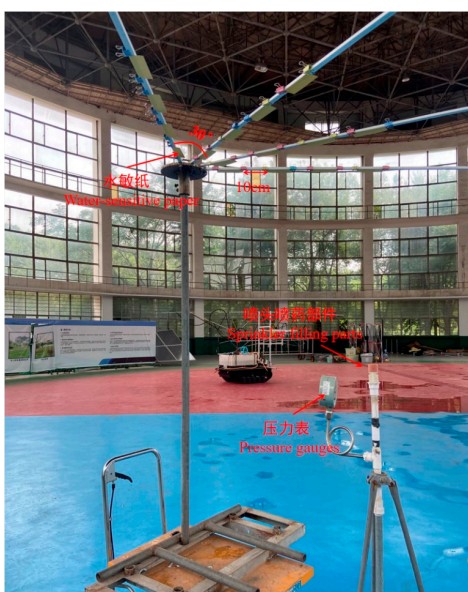

**Figure 5.** Layout of spraying pesticide performance.

Water sensitive paper was used to detect the deposition and droplet distribution during the performance tests of pesticide spraying. The grape leaf surface height of the common trellis is 2 m [23] and the installation height of the sprinkler is 1.2 m. Therefore, the data from grape leaves 80 cm above the sprinkler was simulated to collect. The center is directly above the sprinkler, four sampling lines were arranged 80 cm above the sprinkler,

the interval of each sampling lines was 30°, and the interval of each water sensitive paper was 10 cm. The spray liquid was water and the pressure was 400 kPa. The water sensitive paper were collected and sealed immediately after test [24]. Droplets performance were analyzed. The information from droplet deposition distribution, such as the middle droplet volume diameter, the relative span of droplet spectrum, and the coverage rate of droplet were obtained. The calculation method was detailed in References [25–29].

## 3. Results and Discussion

### 3.1. Influence of Different Structural Parameters on Irrigation Performance

Table 1 shows the results of the irrigation performance and the flow rate of each sprinkle fluctuates 0.75~0.78 m³/h.

**Table 1.** The results of irrigation performance.

| Serial Number | Diversion Hole Inclination Angle $\alpha$/° | Refractive Cone Length $l$/mm | Refraction Cone Angle $\theta$/° | Cone Hole Distance/mm | Wetted Radius/m | Average Irrigation Application Rate/(mm/h) | Uniformity Coefficient/% |
|---|---|---|---|---|---|---|---|
| 1 | 10 | 7 | 120 | 2.5 | 2.93 | 0.56 | 83.10 |
| 2 | 20 | 7 | 120 | 2.5 | 3.54 | 0.50 | 86.28 |
| 3 | 10 | 11 | 120 | 2.5 | 2.53 | 0.61 | 87.13 |
| 4 | 20 | 11 | 120 | 2.5 | 2.85 | 0.60 | 85.58 |
| 5 | 10 | 7 | 140 | 2.5 | 3.13 | 0.69 | 80.45 |
| 6 | 20 | 7 | 140 | 2.5 | 3.18 | 0.58 | 86.76 |
| 7 | 10 | 11 | 140 | 2.5 | 3.03 | 0.73 | 75.87 |
| 8 | 20 | 11 | 140 | 2.5 | 3.10 | 0.78 | 74.93 |
| 9 | 10 | 7 | 120 | 4.5 | 3.80 | 0.49 | 83.99 |
| 10 | 20 | 7 | 120 | 4.5 | 3.70 | 0.48 | 85.26 |
| 11 | 10 | 11 | 120 | 4.5 | 3.28 | 0.55 | 89.91 |
| 12 | 20 | 11 | 120 | 4.5 | 3.26 | 0.65 | 82.02 |
| 13 | 10 | 7 | 140 | 4.5 | 3.60 | 0.49 | 82.26 |
| 14 | 20 | 7 | 140 | 4.5 | 3.50 | 0.47 | 84.16 |
| 15 | 10 | 11 | 140 | 4.5 | 3.68 | 0.51 | 77.37 |
| 16 | 20 | 11 | 140 | 4.5 | 3.52 | 0.70 | 72.99 |
| 17 | 5 | 9 | 130 | 3.5 | 3.53 | 0.57 | 82.50 |
| 18 | 25 | 9 | 130 | 3.5 | 3.83 | 0.56 | 85.26 |
| 19 | 15 | 5 | 130 | 3.5 | 3.30 | 0.47 | 84.08 |
| 20 | 15 | 13 | 130 | 3.5 | 2.58 | 0.59 | 80.72 |
| 21 | 15 | 9 | 110 | 3.5 | 3.13 | 0.68 | 86.39 |
| 22 | 15 | 9 | 150 | 3.5 | 3.21 | 0.87 | 75.64 |
| 23 | 15 | 9 | 130 | 1.5 | 2.55 | 0.58 | 84.13 |
| 24 | 15 | 9 | 130 | 5.5 | 3.53 | 0.40 | 81.25 |
| 25 | 15 | 9 | 130 | 3.5 | 3.10 | 0.72 | 88.21 |
| 26 | 15 | 9 | 130 | 3.5 | 3.10 | 0.72 | 88.21 |
| 27 | 15 | 9 | 130 | 3.5 | 3.10 | 0.72 | 88.21 |
| 28 | 15 | 9 | 130 | 3.5 | 3.10 | 0.72 | 88.21 |
| 29 | 15 | 9 | 130 | 3.5 | 3.10 | 0.72 | 88.21 |
| 30 | 15 | 9 | 130 | 3.5 | 3.10 | 0.72 | 88.21 |
| 31 | 15 | 9 | 130 | 3.5 | 3.10 | 0.72 | 88.21 |

### 3.1.1. The Response Surface of Variance and the Regression Analysis

The analysis of variance (ANOVA) was used to analyze the influence of different structural parameters on sprinkler performance [30]. Wetted radius, average irrigation application rate, and uniformity coefficient were analyzed. The ANOVA of the uniformity coefficient was shown in Table 2. Where Adj SS is the sum of the squares of deviation from the mean, Adj MS is mean square, F represents the significance of the whole fitted equation, and $p$ is an index to measure the difference between the control group and the experimental group.

The ANOVA of the response regression determines whether the influence of each item on the response variable y is significant through the range of the model $p$. If $p \leq 0.05$, it shows that the influence of this term on the response variable y is significant. If $p > 0.05$, it shows that the influence of this term on the response variable y is insignificant [31]. As shown in Table 3, model $p \leq 0.01$ indicates that the regression equation established by the response regression model has a good fit.

**Table 2.** ANOVA of uniformity coefficient.

| Project | Freedom | Adj SS | Adj MS | F | $p$ |
|---|---|---|---|---|---|
| Model | 14 | 601.618 | 42.973 | 65.21 | 0.000 |
| Linear | 4 | 258.748 | 64.687 | 98.15 | 0.000 |
| $\alpha$ | 1 | 1.307 | 1.307 | 1.98 | 0.178 |
| $l$ | 1 | 39.117 | 39.117 | 59.35 | 0.000 |
| $\theta$ | 1 | 216.961 | 216.961 | 329.21 | 0.000 |
| $h$ | 1 | 1.363 | 1.363 | 2.07 | 0.170 |
| Suqare | 4 | 184.974 | 46.243 | 70.17 | 0.000 |
| $\alpha$ | 1 | 33.084 | 33.084 | 50.20 | 0.000 |
| $l$ | 1 | 61.635 | 61.635 | 93.52 | 0.000 |
| $\theta$ | 1 | 91.816 | 91.816 | 139.32 | 0.000 |
| $h$ | 1 | 53.916 | 53.916 | 81.81 | 0.000 |
| Two-factor interaction | 6 | 157.896 | 26.316 | 39.93 | 0.000 |
| $\alpha \times l$ | 1 | 39.816 | 39.816 | 60.42 | 0.000 |
| $\alpha \times \theta$ | 1 | 2.031 | 2.031 | 3.08 | 0.098 |
| $\alpha \times h$ | 1 | 12.110 | 12.110 | 18.38 | 0.001 |
| $l \times \theta$ | 1 | 103.327 | 103.327 | 156.79 | 0.000 |
| $l \times h$ | 1 | 0.221 | 0.221 | 0.34 | 0.571 |
| $\theta \times h$ | 1 | 0.391 | 0.391 | 0.59 | 0.453 |
| Error | 16 | 10.545 | 0.659 | | |

The $p$ values of $l$, $\theta$, $\alpha^2$, $l^2$, $\theta^2$, $h^2$, $\alpha \times \theta$, $\alpha \times h$, and $l \times \theta$ are less than 0.05, which belonged to the significant influencing factors. Removing insignificant terms, the response regression model of each influencing factor to the uniformity coefficient was calculated as follows:

$$Cu = -405 + 3.366\alpha + 24.85l + 5.502\theta + 11.98h - 0.04302\alpha^2 - 0.367l^2 - 0.01792\theta^2 - 1.373h^2 - 0.1578\alpha \times l - 0.1740\alpha \times h - 0.1271l \times \theta \tag{1}$$

Similarly, the wetted radius and average irrigation application rate were analyzed; this analysis process was the same as the uniformity coefficient. The factors with little influence were ignored; the regression equations for the wetted radius and average irrigation application intensity were obtained.

$$r = 7.56 + 0.0375\alpha - 0.875l - 0.0334\theta + 0.5185h + 0.006334\alpha^2 - 0.001188\alpha \times \theta - 0.01787\alpha \times h + 0.0061l \times \theta \tag{2}$$

$$\overline{\rho} = -0.766 + 0.00418\alpha + 0.2025l - 0.0151\theta + 0.7240h - 0.001633\alpha^2 - 0.01265l^2 + 0.000115\theta^2 - 0.06058h^2 + 0.003208\alpha \times l + 0.00485\alpha \times h - 0.003207\theta \times h \tag{3}$$

where $r$ is the wetted radius, m, $\overline{\rho}$ is the average irrigation application rate, mm/h.

The maximum wetted radius, average irrigation application rate, and uniformity coefficient were taken as the target value; the optimal parameter combination of the key hydraulic structures were obtained by multi-response optimization [32]. The structures values were: the diversion hole inclination angle was 20.8°, the refractive cone angle was 123.7°, the refractive cone length was 8.8 mm, and the cone hole distance was 3.6 mm. The hydraulic performance of the sprinkler was: the wetted radius was 3.4 m, the average irrigation application rate was 0.65 mm/h, and the uniformity coefficient was 88%.

**Table 3.** The results of spraying pesticide performance.

| Test Number | Diversion Chute Width $b$/mm | Number of Diversion Chutes $n$ | Diversion Chute Inclination Angle $\beta$/° | Rotational Acceleration Chamber Height $h_1$/mm | Length of Cylindrical Section at Nozzle Outlet $h_2$/mm | Droplet Volume Mid-Diameter/μm | Droplet Spectral Width | Droplet Coverage/% |
|---|---|---|---|---|---|---|---|---|
| 1 | 1.0 | 2 | 10 | 2 | 0.5 | 289.36 | 1.93 | 7.49 |
| 2 | 2.0 | 2 | 10 | 2 | 0.1 | 307.11 | 2.52 | 7.03 |
| 3 | 1.0 | 4 | 10 | 2 | 0.1 | 289.3 | 4.13 | 6.40 |
| 4 | 2.0 | 4 | 10 | 2 | 0.5 | 127.17 | 1.94 | 2.97 |
| 5 | 1.0 | 2 | 30 | 2 | 0.1 | 240.17 | 1.44 | 2.21 |
| 6 | 2.0 | 2 | 30 | 2 | 0.5 | 315.22 | 1.96 | 7.66 |
| 7 | 1.0 | 4 | 30 | 2 | 0.5 | 309.03 | 1.41 | 6.78 |
| 8 | 2.0 | 4 | 30 | 2 | 0.1 | 231.57 | 3.59 | 6.78 |
| 9 | 1.0 | 2 | 10 | 4 | 0.1 | 245.78 | 3.30 | 7.91 |
| 10 | 2.0 | 2 | 10 | 4 | 0.5 | 172.76 | 2.85 | 4.71 |
| 11 | 1.0 | 4 | 10 | 4 | 0.5 | 300.52 | 2.87 | 6.01 |
| 12 | 2.0 | 4 | 10 | 4 | 0.1 | 241.11 | 4.08 | 11.77 |
| 13 | 1.0 | 2 | 30 | 4 | 0.5 | 227.72 | 1.57 | 1.89 |
| 14 | 2.0 | 2 | 30 | 4 | 0.1 | 295.92 | 2.83 | 8.87 |
| 15 | 1.0 | 4 | 30 | 4 | 0.1 | 250.56 | 2.36 | 11.81 |
| 16 | 2.0 | 4 | 30 | 4 | 0.5 | 188.98 | 2.54 | 4.01 |
| 17 | 0.5 | 3 | 20 | 3 | 0.3 | 248.49 | 1.24 | 3.22 |
| 18 | 2.5 | 3 | 20 | 3 | 0.3 | 157.81 | 2.65 | 3.03 |
| 19 | 1.5 | 1 | 20 | 3 | 0.3 | 272.51 | 1.32 | 6.19 |
| 20 | 1.5 | 5 | 20 | 3 | 0.3 | 201.30 | 3.23 | 6.15 |
| 21 | 1.5 | 3 | 0 | 3 | 0.3 | 229.30 | 3.48 | 7.55 |
| 22 | 1.5 | 3 | 40 | 3 | 0.3 | 251.64 | 1.12 | 4.86 |
| 23 | 1.5 | 3 | 20 | 1 | 0.3 | 284.64 | 1.62 | 5.24 |
| 24 | 1.5 | 3 | 20 | 5 | 0.3 | 240.93 | 2.20 | 7.19 |
| 25 | 1.5 | 3 | 20 | 3 | 0.1 | 266.32 | 3.06 | 7.87 |
| 26 | 1.5 | 3 | 20 | 3 | 0.7 | 322.12 | 2.19 | 7.33 |
| 27 | 1.5 | 3 | 20 | 3 | 0.3 | 175.51 | 2.50 | 6.01 |
| 28 | 1.5 | 3 | 20 | 3 | 0.3 | 175.51 | 2.50 | 6.01 |
| 29 | 1.5 | 3 | 20 | 3 | 0.3 | 175.51 | 2.5 | 6.01 |
| 30 | 1.5 | 3 | 20 | 3 | 0.3 | 175.51 | 2.5 | 6.01 |
| 31 | 1.5 | 3 | 20 | 3 | 0.3 | 175.51 | 2.5 | 6.01 |

### 3.1.2. Analysis of Main Effect and the Interaction Effect

Figure 6 shows the main effect plot of irrigation performance.

As shown in Figure 6a, the wetted radius increased first and then decreased as the refractive cone angle increased. It is negatively correlated with the refractive cone length and positively correlated with the cone hole distance. The refractive cone angle is less insignificant to the wetted radius than other factors. As shown in Figure 6b, average application rate has a quadratic function relationship with the diversion hole inclination angle, the refractive cone length, and the cone hole distance, which increases slowly with the increase in the refractive cone angle. As shown in Figure 6c, with each structural parameter, the uniformity coefficient initially shows an increasing and then decreasing trend.

Figures 7–9 show the two-factor interaction diagram of the response variable between the wetted radius, average application rate, uniformity coefficient, diversion hole inclination angle, refractive cone angle, refractive cone length, and cone hole distance. The contour map is the projection of the response surface on the horizontal plane. Each group of two-factor interaction diagrams is the interaction between two independent variables when other variables are regarded as zero level.

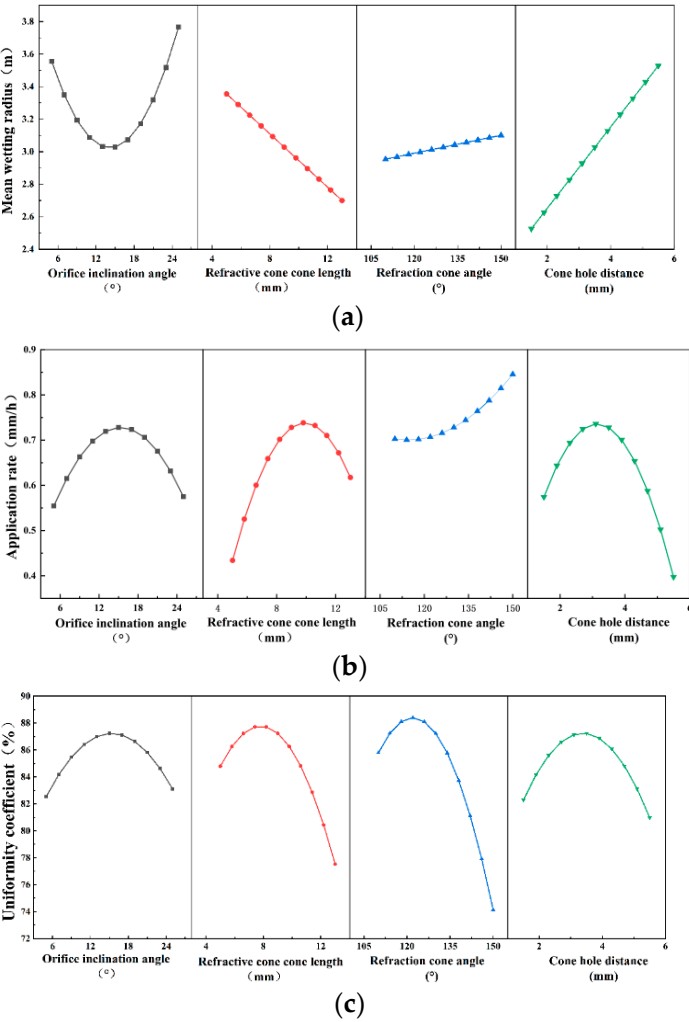

**Figure 6.** Main effect plot of irrigation performance. (**a**) Influencing factors of the wetted radius, (**b**) influencing factors of sprinkler application rate, (**c**) influencing factors of uniformity coefficients.

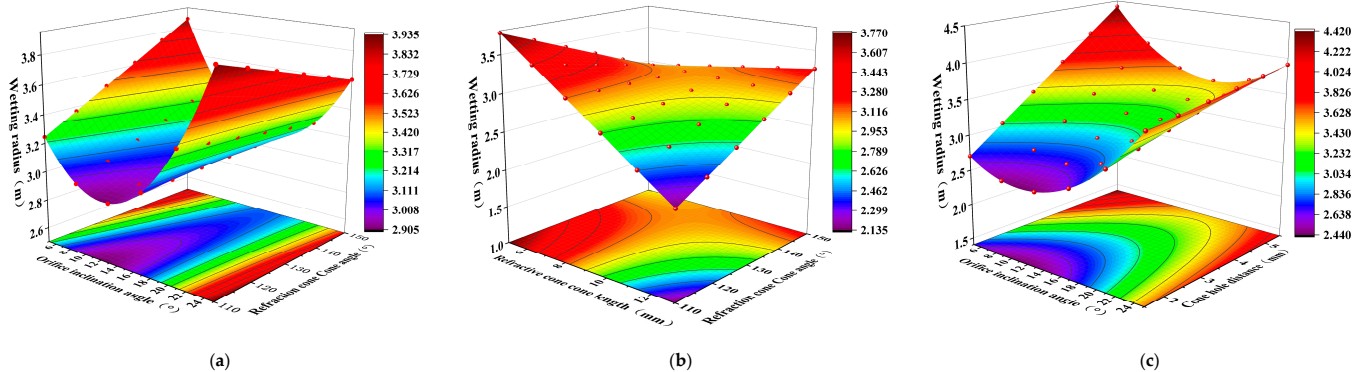

**Figure 7.** Two-factor interaction effect diagram of wetted radius. (**a**) Interaction effect between $\alpha$ and $\theta$, (**b**) interaction effect between $l$ and $\theta$, (**c**) interaction effect between $\alpha$ and $h$.

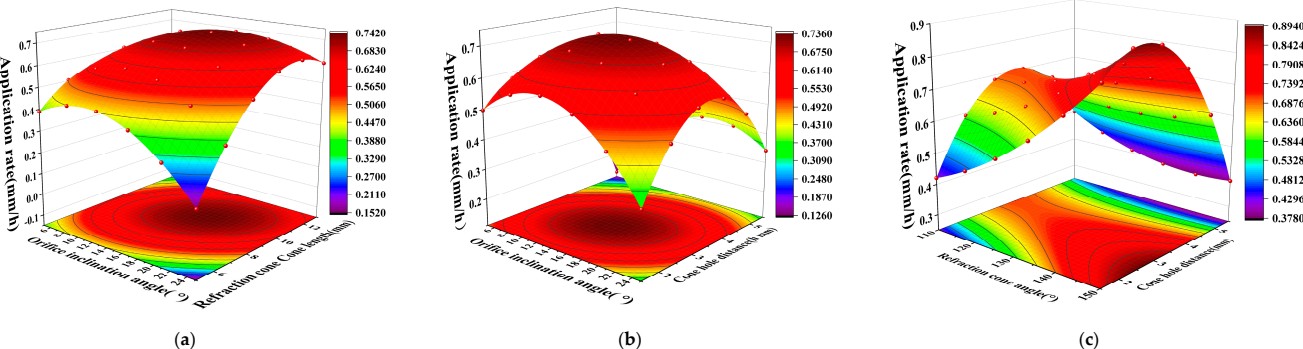

**Figure 8.** Two-factor interaction effect diagram of application rate. (**a**) Interaction effect between $\alpha$ and $l$. (**b**) Interaction effect between $\alpha$ and $h$. (**c**) Interaction effect between $\theta$ and $h$.

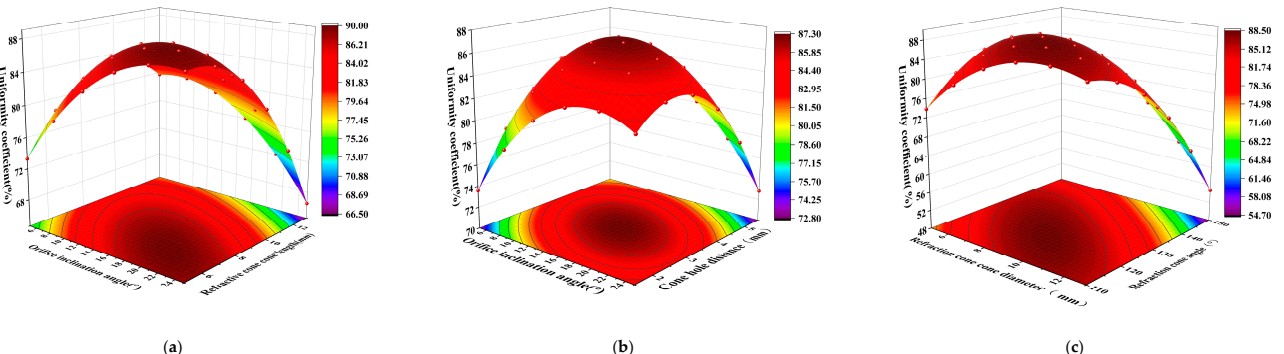

**Figure 9.** Two-factor interaction effect diagram of uniformity coefficients. (**a**) Interaction effect between $\alpha$ and $l$. (**b**) Interaction effect between $\alpha$ and $h$, (**c**) Interaction effect between $l$ and $\theta$.

As shown in Figure 7a,c, the interaction between the diversion hole inclination angle, the refractive cone angle, and the cone hole distance is significant. When the diversion hole inclination angle range is (5°~18°), the refractive cone angle is negatively correlated with the wetted radius within the range. At the upper limit of the cone hole distance (5.5 mm), the maximum wetted radius is 4.42 m. As shown in Figure 7b, when the refractive cone angle is (110°~140°), the wetted radius decreases with the increase in the refractive cone length, and the range gradient of the decreasing process is large, indicating that the refractive cone length has a significant influence on the wetted radius.

As shown in Figure 8a,b, the average irrigation application intensity increases first and then decreases as the diversion hole inclination angle and the refractive cone length increase, inside which the area of high efficiency is large. Figure 8c shows that, at upper limit of the refractive cone angle (150°) and when the range of the cone hole distance is 2.5~3.5 mm, the maximum average irrigation application intensity is 0.894 mm/h.

As shown in Figure 9a,b, when the diversion hole inclination angle is 25°, the uniformity coefficient changes obviously with the refractive cone length, which shows that the refractive cone length has a significant influence on the uniformity coefficient; the uniformity coefficient increases first and then decreases as the diversion hole inclination angle and the cone hole distance increase. When the range of cone hole distance is 2.5~3.5 mm and the range of the diversion hole inclination angle is 13°~21°, the uniformity coefficients reach the maximum value of 87.3%. As shown in Figure 9c, the contour plot shows that under the interaction between the diversion hole inclination angle and refractive cone length, the high-efficiency area of uniformity coefficient is more significant. When the range of the refractive cone angle is (110°~115°) and the refractive cone length is (9~12 mm), the uniformity coefficients reach the maximum value.

### 3.2. Influence of Different Structural Parameters on Spraying Pesticide Performance

Table 3 shows the results of the performance of spraying pesticide and the flow rate of each sprinkler fluctuates 0.15~0.2 m$^3$/h.

#### 3.2.1. The Response Surface of Variance and the Regression Analysis

Droplet volume mid-diameter, droplet spectral width, and droplet coverage were analyzed. The ANOVA of the droplet volume mid-diameter was shown in Table 4.

**Table 4.** *ANOVA* of droplet volume mid-diameter.

| Project | Freedom | Adj SS | Adj MS | F | p |
|---|---|---|---|---|---|
| Model | 20 | 86,113.0 | 4305.6 | 11.12 | 0.000 |
| Linear | 5 | 16,156.0 | 3652.1 | 9.43 | 0.001 |
| $b$ | 1 | 8587.4 | 8587.4 | 22.18 | 0.001 |
| $n$ | 1 | 3706.4 | 3706.4 | 9.57 | 0.010 |
| $\beta$ | 1 | 712.2 | 712.2 | 1.84 | 0.202 |
| $h_1$ | 1 | 3105.3 | 3105.3 | 8.02 | 0.016 |
| $h_2$ | 1 | 44.7 | 2149.3 | 5.55 | 0.038 |
| Suqare | 5 | 36,060.3 | 7212.1 | 18.63 | 0.000 |
| $b \times b$ | 1 | 6.4 | 77.2 | 0.20 | 0.664 |
| $n \times n$ | 1 | 2182.9 | 2982.6 | 7.70 | 0.018 |
| $\beta \times \beta$ | 1 | 3077.9 | 3535.6 | 9.13 | 0.012 |
| $h_1 \times h_1$ | 1 | 8622.0 | 8055.2 | 20.81 | 0.001 |
| $h_2 \times h_2$ | 1 | 22,171.1 | 22,171.1 | 57.27 | 0.000 |
| Two-factor interaction | 10 | 33,896.7 | 3389.7 | 8.76 | 0.001 |
| $b \times n$ | 1 | 12,574.9 | 12,574.9 | 32.48 | 0.000 |
| $b \times \beta$ | 1 | 4935.8 | 4935.8 | 12.75 | 0.004 |
| $b \times h_1$ | 1 | 27.5 | 27.5 | 0.07 | 0.795 |
| $b \times h_2$ | 1 | 8668.1 | 8668.1 | 22.39 | 0.001 |
| $n \times \beta$ | 1 | 110.1 | 110.1 | 0.28 | 0.604 |
| $n \times h_1$ | 1 | 3415.6 | 3415.6 | 8.82 | 0.013 |
| $n \times h_2$ | 1 | 0.5 | 0.5 | 0.00 | 0.971 |
| $\beta \times h_1$ | 1 | 400.4 | 400.4 | 1.03 | 0.331 |
| $\beta \times h_2$ | 1 | 2922.3 | 2922.3 | 7.55 | 0.019 |
| $h_1 \times h_2$ | 1 | 841.4 | 841.4 | 2.17 | 0.168 |
| Error | 11 | 4258.4 | 387.1 | | |

The following regression equations for the droplet volume mid-diameter, droplet spectral width, and droplet coverage were obtained.

$$Dv_{0.5} = 599.5 + 129.9b - 31.7n - 11.08\beta - 153.6h_1 - 375h_2 + 9.93n \times n + 0.1082\beta \times \beta + 16.40h_1 \times h_1 + 892\,h_2 \times h_2 \\ - 56.07b \times n + 3.513b \times \beta - 232.8b \times h_2 + 14.61n \times h_1 + 6.76\beta \times h_2 \tag{4}$$

$$S = 2.891 - 0.732b + 0.687n - 0.1377\beta + 0.1928h_1 - 2.53h_2 + 5.98h_2 \times h_2 + 0.0622b \times \beta - 1.131n \times h_2 \tag{5}$$

$$Deposits = -7.73 + 14.19b + 0.943n - 0.3156\beta + 0.839h_1 + 23.39h_2 - 2.806b \times b + 25.01h_2 \times h_2 - 1.777b \times n + \\ 0.0742b \times \beta - 5.59b \times h_2 + 0.0546n \times \beta + 0.728n \times h_1 - 3.979n \times h_2 - 8.196\,h_1 \times h_2 \tag{6}$$

where $Dv_{0.5}$ is droplet volume mid-diameter, μm, $S$ is droplet spectral width, and *Deposits* is droplet coverage, %.

The droplet volume medium diameter of 200 μm, the droplet spectral width of 1, and the droplet coverage were taken as the target value. The structures values were: the diversion chute width was 2.5 mm, the number of diversion chutes was 2, the diversion chute inclination angle was 10°, the rotary acceleration chamber height was 1.3 mm, and the nozzle outlet cylindrical section length was 0.7 mm. The droplet volume mid-diameter was 200.2 μm, the droplet spectrum width was 2.2, and the droplet coverage was 9.4%.

### 3.2.2. Analysis of Main Effect and the Interaction Effect

Figure 10 shows the main effects diagram of the diversion chute width, the number of diversion chutes, the diversion chute inclination angle, the rotary acceleration chamber height, the nozzle outlet cylindrical section length on the droplet volume mid-diameter, droplet spectral width, and droplet coverage.

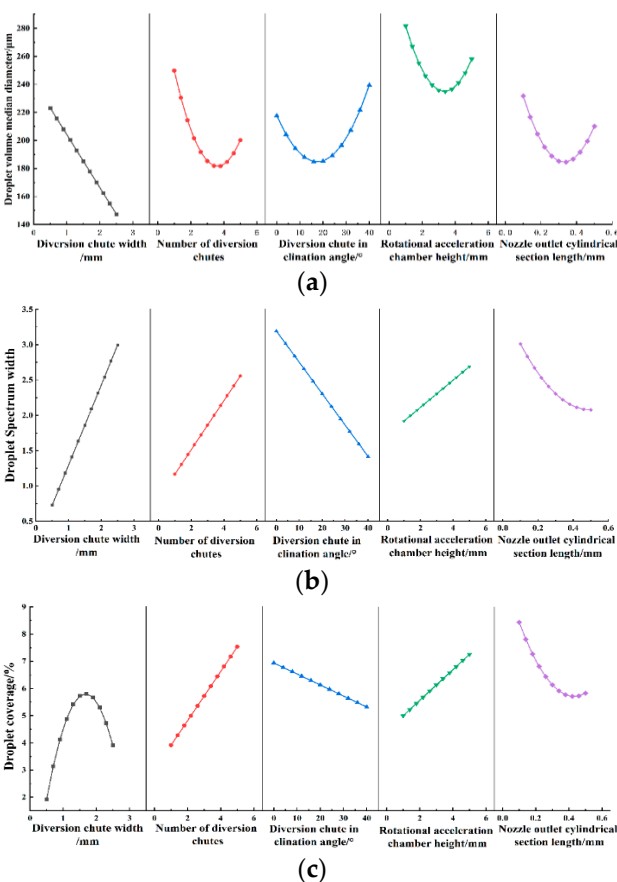

**Figure 10.** Main effect plot of spraying pesticide. (**a**) Influencing factors of droplet volume medium diameter, (**b**) influencing factors of droplet spectrum width, (**c**) influencing factors of droplet coverage.

As shown in Figure 10a, the droplet volume mid-diameter is negatively correlated with the diversion chute width; the other factors have a quadratic function relationship with the volume diameter of the spray head, and the wave troughs exist. As shown in Figure 10b, the droplet spectral width is positively corrected with the diversion chute width, the number of diversion chutes n, and the rotary acceleration chamber height. Moreover, the droplet spectral width is negatively corrected with the diversion chute inclination angle and the nozzle outlet cylindrical section length. As shown in Figure 10c, the droplet coverage increases as the number of diversion chutes and the rotary acceleration chamber height increase.

Figures 11–13 show the two-factor interaction diagram of the response variable between droplet volume diameter, droplet spectral width, droplet coverage and the diversion chute width, the number of diversion chutes, the diversion chute inclination angle, the rotary acceleration chamber height, and the nozzle outlet cylindrical section length.

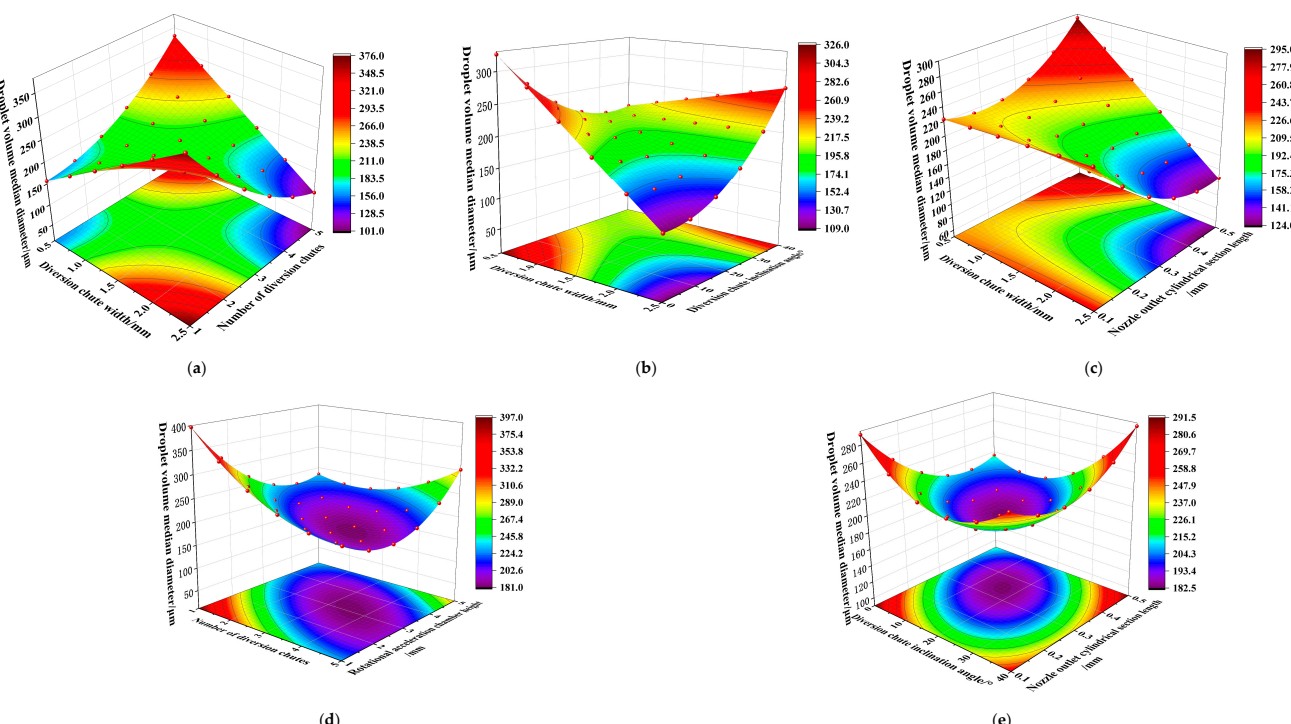

**Figure 11.** Two-factor interaction effect diagram of the droplet volume medium diameter. (**a**) Interaction effect between *b* and *n*, (**b**) interaction effect between *b* and *β*, (**c**) interaction effect between *b* and $h_2$, (**d**) interaction effect between *n* and $h_1$, (**e**) interaction effect between *β* and $h_2$.

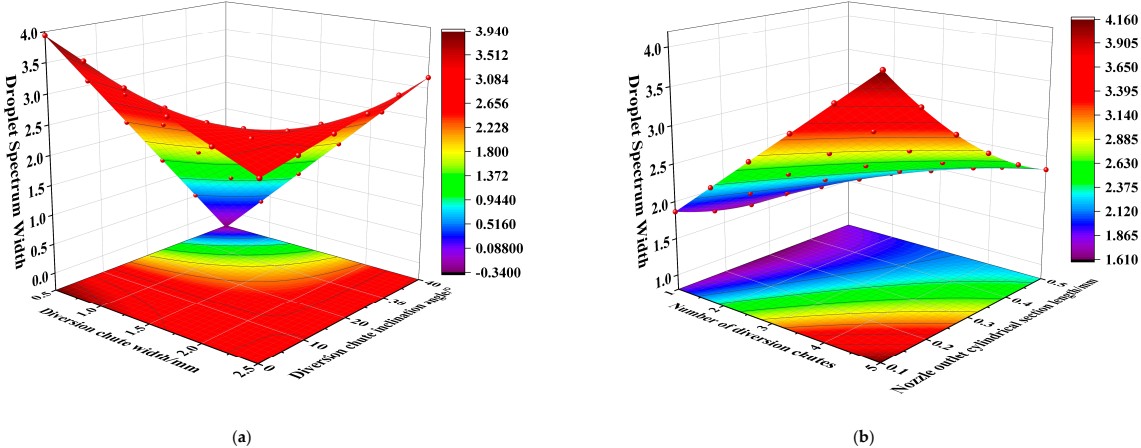

**Figure 12.** Two-factor interaction effect diagram of the droplet spectrum width. (**a**) Interaction effect between *b* and *β*, (**b**) interaction effect between *n* and $h_2$.

As shown in Figure 11a–c, under the interaction effect between the diversion chute width and the diversion chute inclination angle, the droplet volume mid-diameter changes more significantly when the diversion chute width is 0.5 mm, the diversion chute inclination angle is 0°, and the droplet volume mid-diameter reaches the maximum value 326 μm; the diversion chute width has a significant influence on the droplet volume mid-diameter when the diversion chute is less than 1.5 mm, and the droplet volume mid-diameter is positively corrected with the nozzle outlet cylindrical section length. This change is the opposite when the diversion chute is greater than 1.5 mm. As shown in Figure 11e, which is the same as Figure 11d, the diversion chute inclination angle and the nozzle outlet cylindrical section length significantly influence the droplet volume mid-diameter.

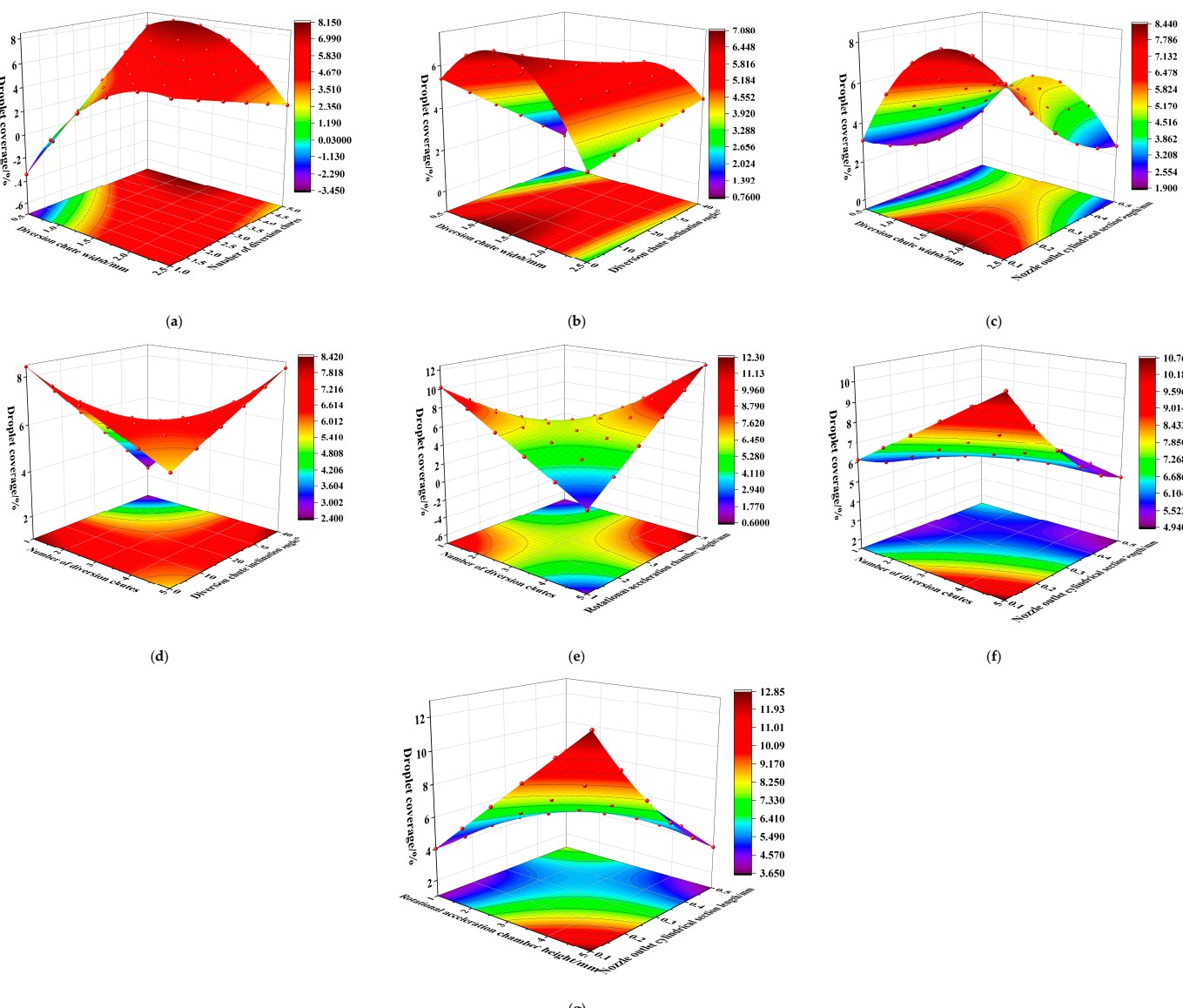

**Figure 13.** Two-factor interaction effect diagram of the droplet coverage. (**a**) Interaction effect between $b$ and $n$, (**b**) interaction effect between $b$ and $\beta$, (**c**) interaction effect between $b$ and $h_2$, (**d**) interaction effect between $n$ and $\beta$, (**e**) interaction effect between $n$ and $h_1$, (**f**) interaction effect between $n$ and $h_2$, (**g**) interaction effect between $h_1$ and $h_2$.

As shown in Figure 12a, the 3D curved surface of the diversion chute width and the diversion chute inclination angle on t response variable droplet spectral width is "saddle-like". When the diversion chute inclination angle is less than 20°, the diversion chute width has an insignificant influence on the droplet spectral width;As shown in Figure 12b, when the nozzle outlet cylindrical section length and the number of diversion chutes are fixed, the droplet spectrum width is positively correlated with the number of diversion chutes.

As shown in Figure 13, for the response variable droplet coverage, there are more significant terms of the two-factor interaction; the diversion chute width significantly influences the droplet coverage. As shown in Figure 13a–c, when the lower limit values are taken separately by the diversion chute width and the number of diversion chutes, the droplet coverage is negative. When the diversion chute width is less than 1.25 mm, the droplet coverage decreases as the diversion chute inclination angle increases. As shown in Figure 13d–f, when the number of diversion chutes are less than three, the droplet coverage is negatively corrected with the rotary acceleration chamber height. Under the interaction

effect between the number of diversion chutes and the nozzle outlet cylindrical section length, the droplet coverage changes are more significant, as the maximum is 10.76% and the minimum is 4.94%. As shown in Figure 13g, when the rotary acceleration chamber height is 5 mm, the nozzle outlet cylindrical section length is 0.1mm and the response surface diagram reaches the maximum value of 12.85%.

## 4. Conclusions

A new water and pesticide integrated sprinkler was designed. The ANOVA and the regression analysis of the response surface were used to analyze the influence of key structural parameters on the performance of a water–pesticide integrated sprinkler. The conclusions were as follows:

(1) The influences of key structural parameters, such as the diversion hole inclination angle, the refractive cone angle, the refractive cone length, and the cone hole distance on sprinkler irrigation performance were revealed. The influence laws of key structural parameters of different diversion chute widths, the number of diversion chutes, the diversion chute inclination angle, the rotary acceleration chamber height, and the nozzle outlet cylindrical section length on sprinkler spraying pesticide performance were revealed, respectively.

(2) Taking the wetted radius, average irrigation application rate, and uniformity coefficient as the parameters of irrigation performance and droplet volume mid-diameter, droplet spectral width, and droplet coverage as the parameters of spraying pesticide performance, the proposed design values of key structural parameters with better performance are obtained: the diversion hole inclination angle was 20.8°, the refractive cone angle was 123.7°, the refractive cone length was 8.8 mm, the cone hole distance was 3.6 mm, the diversion chute width was 2.5 mm, the number of diversion chutes was 2, the diversion chute inclination angle was 10°, the rotary acceleration chamber height was 1.3 mm, and the nozzle outlet cylindrical section length was 0.7 mm. The performance parameters of the sprinkler can meet the needs of grape irrigation and pesticide spraying, thereby improving the utilization rate of water and pesticides in the agricultural production process.

**Author Contributions:** Data curation, X.W.; validation, Q.L. and X.W.; formal analysis, J.L.; resources, J.L.; funding acquisition, J.L.; investigation, Y.Z. and Z.H.; supervision, J.L.; writing—original draft, X.W.; writing—review and editing, Z.H. All authors have read and agreed to the published version of the manuscript.

**Funding:** This research was funded by the Key R&D plan project of Jiangsu Province (No. BE2021341), Project of Faculty of Agricultural Equipment of Jiangsu University (No. NZXB20210101), and Postgraduate Research Practice lnnovation of Jiangsu Province (SJCX22_1868).

**Conflicts of Interest:** The authors declare no conflict of interest.

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
