# Peer review of "Optimization of Key Hydraulic Structure Parameters of a New Type of Water–Pesticide Integrated Sprinkler Based on Response Surface Experiment"

_water, doi:10.3390/w15081486_

Round 1

Reviewer 1 Report

1. 31 groups of response surface tests were designed for sprinkler irrigation test and pesticide spraying test. But the number of sprinkler components in the result table is 25 and 27 respectively. Is there a contradiction here? It is suggested to add references to the experimental design method of response surface.

2. The duration of spraying is not mentioned in "2.2 Test conditions and scheme", but is required for the calculation of the analytical indicators selected for the subsequent analysis of the results and is recommended to be added.

3. Section 3.1 lack descriptions of the process of obtaining the three indicators of wetting radius, average spraying intensity and uniformity coefficient, such as calculation formulae, etc. It is recommended that these be added.

4. Section 3.2.1 lack ANOVA table on spraying pesticide performance. It is recommended that these be added.

Author Response

The authors greatly appreciate the valuable and insightful comments made by the reviewers and would like to thank them for their efforts. Their comments have undoubtedly helped us to improve the quality of our manuscript. We have carefully considered all the comments and the manuscript has been revised thoroughly to meet their expectations. Our point-by-point responses to each comment. Please see the attachment.

Reviewer 2 Report

The study is a well-designed experiment to develop a sprinkled system that improves the simultaneous application of irrigation water and pesticide to grape vines. The study includes the testing of the most important hydraulic parameters of such system and applies sound statistical methods to analyze their performance.

The article is clearly written and well organized, and Tables and figures are of good quality. However, all of text, figures and tables require some improvement. A general statement follows and a detailed comments are included below.

General statement: The main reason for this work was to find a way to both irrigate and apply pesticide in a more effective manner, but this is mentioned only at the beginning of the paper. The paper then proceeds with the engineering specifications of the work, and the relation of the work to the main objectives is not mentioned anymore. These need to be added to results and conclusions, to state clearly how the results obtained in this study relate to the main objectives.

Specific comments:

Remove sentence in lines 19-21

Include a sentence at the end of the abstract indicating if the goals of the study were met with respect to adding efficiency to the cultivation of grape vines.

Line 34. Sentence seems out of place for a technical paper, change to something like “In recent years, China has increased its fruit production significantly”.

Lines 37-39 are unclear. Please add clarity. It seems a better word to mainly (line 37) would be either, and the beginning of the sentence (line 38) should be these and not this, or if you are referring to the sprayer method, then the sentence should start with “The latter…”

Lines 38-39. What do you mean by large drift losses? Somehow the difference between “large drift losses” and “low pesticide utilization” is not made clear. Please explain more clearly.

Lines 40-41. The first sentence of this paragraph reads odd. Maybe start with “An effective way to solve the problem….. is to combine the functions of…..

Line 44. Remove space between root and period. Add space between any number and its units (throughout all manuscript).

Line 51-52. “…trees include centrifugal” ….. and end the sentence with “among others”

Line 52 correct spelling of sprinklers (first word in this line)

Line 53. Replace “can” for “do”

Line 60. … sprinkler, in which the influence…

Line 61.  … studied. Their results provided a theoretical…

Lines 63-66. This sentence needs clarity, please rewrite

Line 71-72 Make this sentence more specific to the expected results… “important basis for the engineering application of”.. does not say anything really, it is ambiguous.

Line 99.  A space is needed between “(a)” and “Structure” in the figure tittle, as well as for “(b)” and “Structure”. Leave one space (empty line) between the Figure title and the text.

Lines 100 and 106: The first is an incomplete sentence. Instead connect to the second sentence, something like “Figure 3(a) shows the sprinkler irrigation…” and “Figure 3(b) shows the structure parameters associated to spraying pesticide: the diversion chute….”

Line 116-117.  replace “was shown” for “is shown”

Line 117. Add a space between “Figure” and “5”

Line 123. Rewrite this sentence (grammar).

Line 147. Sentence is not clear. Please rewrite. Maybe listing the random variables right after the term is mentioned? What do you mean by mathematical statistics?

Line 150. Tables need to be self-explanatory. Spell out each of the terms in the top row, correct the word “source”. Leave a space (empty row) between bottom of the table and the text.

Lines 151-158. This paragraph is not a result, it should be moved to the methods section.

Lines 180-185.  Increase font size of axis labels and add a space between “(a)” and “Influencing:. Repeat for (b) and (c). Leave a space (empty row) between bottom of figure and text. Correct all figures (font size, spaces)

Line 341. Complete first sentence by adding the goals of this study … sprinkler was designed to (add goals here) …..

Line 366. Add a sentence with the significance of these results to the production of grape vine and the protection of resources, maybe prevention to the contamination of orchard environment?

Author Response

(The authors gave the same response as above.)
